# Review of the Effects of Developments with Low Parking Requirements

**Frances Sprei [1],\*, Cecilia Hult [2], Åsa Hult [2] and Anders Roth [2]**

[1]   Space, Earth & Environment, Chalmers University of Technology, 412 96 Gothenburg, Sweden
[2]   IVL, Swedish Environmental Research Institute, 411 33 Gothenburg, Sweden; cecilia.hult@ivl.se (C.H.);
     asa.hult@ivl.se (A.H.); anders.roth@ivl.se (A.R.)
\*   Correspondence: frances.sprei@chalmers.se; Tel.: +46-31-7722146

**Abstract:** Parking management and planning can be used to address several issues related to sustainable urban development. For example, parking availability affects both car ownership and usage, and parking planning can affect both land use and building costs. A tool used in several countries is minimum parking requirements (MPR) and lowering these could be a pathway to more sustainable mobility. However, the actual effects of lower MPR have not systematically been studied. In this paper we present the results of a review of sixteen developments with low MPR in Sweden, Austria, Germany, Switzerland, and the UK. Existing research and reports have been analyzed to compare these and draw conclusions on the effect of MPR on mobility patterns and mobility services. In addition, interviews were conducted with representatives from municipalities and developers. Our results indicate that the mobility patterns of individuals in the studied projects are more sustainable than in nearby projects. However, the causality of MPR and mobility is hard to establish due to the risk of self-selection and that all of the studied projects have good prerequisites for sustainable mobility practices. Many of the studied evaluations are also of poor quality with, for example, lack of appropriate control group.

**Keywords:** minimum parking requirements; carsharing; mobility patterns; car ownership

---

## 1. Introduction

Faced with increasing challenges with global (e.g., climate change) as well as local implications (air quality and land use), cities are looking at new ways to address issues such as transportation planning and urban land use [1]. One core issue is to move away from the status quo of the car being the dominant mode of transport [2]. In a study of travel behavior with regard to mode of transport and trip purpose (work, shopping, and leisure), car ownership was the most important variable—if people own a car, they use it [3]. There is also a link between vehicle ownership, use of vehicles, and parking availability [4,5]. Thus, reducing parking availability can have beneficial environmental consequences but might imply less mobility and accessibility for people. Combining restrictions on parking with access to mobility services such as carsharing may be an attractive solution to this dilemma [6].

Another strategy to shift mobility patterns is to incentivize the use of alternative modes of transport. Studies in the US, however, have shown that the effect of these incentives is strongly reduced by free workplace parking [7]. Christiansen et al [8] found, in Norway, that the availability of both residential parking and workplace parking affect vehicle use. There is also growing evidence for the connection between the availability of parking and increased car ownership and car use, even in areas with good access to public transport [4,5,9,10]. A review report from the Swedish Transport Administration found that several studies point to parking prices as an efficient tool to reduce congestion and vehicle use in cities [11].

Urban planners have been setting minimum parking requirements (MPR) for different types of land use. Shoup [12] lists different types of land use that have, at least in the US, been subjected to minimum parking requirements and these include bingo parlors, veterinarians, and convents. Most commonly minimum parking requirements have been associated with residential buildings, offices, and other commercial developments. In Sweden, they have been used for housing from the 1950s [13]. Minimum parking requirements related to residential buildings are normally formulated in relation to the number of apartments, such as one parking space per apartment. In Sweden they were set based on US conditions, despite car dependency being much larger in the US. The actual origin or motivational grounds for minimum parking requirements is not very clear [12]. It thus seems that parking has for a long time been regulated on little empirical evidence of its effects and weak scientific base for its implementation.

Increasing evidence that liberal parking requirements lead to more expensive housing prices and to increased car use [4,5,9,14] has led to a redesigning of the requirements. New parking management strategies have been recently used by several cities in Europe to manage mobility, congestion, and to create a more attractive environment [15]. For residential buildings, flexible parking requirements often allow for reductions in the number of parking places provided other solutions, such as carsharing, are available [16]. In Sweden, over 30 cities have introduced lower parking requirements if mobility services are provided. Lower parking requirements exist in the major Swedish cities of Stockholm, Gothenburg, and Malmö, as well as smaller cities. Internationally, the cities of Washington, Portland, London, Berlin, and San Francisco have adopted flexible parking requirements. In London, for example, there are extensive controlled parking zones that make vehicle ownership very cumbersome [17].

Melia [18] has studied various car free and car limited developments. There are many areas in the world that are car free because no road access is possible (the most famous example being Venice), but the focus of his work and this paper is on developments where there is deliberate choice in reducing car access and ownership. In this paper we delimited this to new areas, and not to changes in existing areas.

The shift to new forms of parking requirements has sparked a need for knowledge, especially in combination with the provision of mobility services. Urban planners and developers need to better understand how these parking norms should be designed and what the consequences are. What is an appropriate level of parking requirement? What factors determine an appropriate level of parking requirement? Do they actually reduce vehicles ownership, or do they transfer vehicles to on-street parking instead? Does low-parking housing mainly attract people that already have a car-free lifestyle? What is the demand for these types of developments? As a step in trying to address these questions, we have studied evaluations of fifteen existing low-parking developments. The aim of this study has been to look at the effect on mobility patterns, try to identify what lessons can be learned by projects so far, but also look at the quality of the evaluations and what recommendations can be made for future studies.

For this reason, we have identified two major research question that are further divided into sub questions:

1) How are these developments evaluated and what lessons can be learned about low parking requirements based on existing evaluations?

   a) What mobility services are provided to reduce car dependency?
   b) Do the evaluations show any evidence of more sustainable mobility patterns? More specifically is private car ownership reduced? And how are the mobility services provided such as carsharing used?
   c) What lessons can be learned from a planning and process perspective?

2) What is the quality of the evaluations?

   a) What type of actor carried out the evaluation?

b)      What methods are used?

c)      Is a control group used to compare the results?

The paper is structured in the following manner. First, we describe the data that has been collected, the selection criteria for the developments and the methods used to analyze the data in the Data and Methods section. In the Results section we present first the results related to the first research question, i.e., summary and lessons learned from the evaluations, and thereafter the results regarding the quality of the evaluations. In the section after Results, we look at developments that have recently been constructed or that are planned. We end with conclusions and recommendations.

## 2. Materials and Methods

We based our analysis on sixteen developments with low(er) minimum parking requirements with the majority (eleven) in Sweden. As a reference, we also selected developments in other European countries: one in Austria, two in Germany, one in Switzerland, and one in the UK. Most of the developments are residential buildings, however, one in Sweden is an office building and the one in Switzerland is a shopping mall and workplace. The reviewed developments can be found in Table 1.

**Table 1.** Overview of studied developments.

| Name of Development | Type of Building | Country |
| --- | --- | --- |
| Almedal | Residential | Sweden |
| Elvsjoe | Residential | Sweden |
| Embla | Commercial offices | Sweden |
| Florisdorf | Residential | Austria |
| Fullriggaren | Residential | Sweden |
| Hammarby Sjöstad | Residential | Sweden |
| Haninge | Residential | Sweden |
| Ohboy | Residential (and hotel) | Sweden |
| Kvillebäcken | Residential | Sweden |
| Pool Quarter | Residential | UK |
| Porslinsfabriken | Residential | Sweden |
| Sihlcity | Commercial (shopping mall and offices) | Switzerland |
| Stellwerk 60 | Residential | Germany |
| Stockholm Royal Seaport | Residential | Sweden |
| Vauban | Residential | Germany |
| Viva | Residential | Sweden |

The study is mainly based on secondary data: literature review of research papers, evaluations and reports from the developments, building permits, and detail plans of the developments. In addition, interviews were made with representatives from municipalities and developers in Sweden.

We selected the developments based on the following criteria:

- They had lower minimum parking requirements than the rest of the municipality, or similar areas in other municipalities.
- They were fully constructed and inhabited.

In addition to this we also identified the following criteria as desirable but not necessary:

- The development had been evaluated in some way regarding effects on car ownership and car usage/mobility behavior (We were not that strict on exactly what type of evaluation had been done, but rather that some kind of assessment of the effects had been made). Evaluations carried out by researchers were prioritized.
- The development included some kind of mobility service such as carsharing.

In the first screening, 32 developments were identified. Sixteen of these met the above criteria and were thus included in the study. Municipalities, organizations with knowledge of parking, and a reference group helped us to identify existing developments in the first step.

The main focus has been developments in Sweden. The eleven selected developments in Sweden are representative for inhabited and evaluated developments. The five included developments from other European countries were chosen since they had been fully or partly inhabited for at least 10 years and had been evaluated. Four out of five European developments have lower parking requirements than the majority of Swedish projects and can possibly be seen as extreme cases even in a European context. For a more in-depth description of the developments, see [19].

Evaluations and other material have been analyzed based on a number of factors. First, more descriptive parameters such as type of development, size, and parking norm (parking space/apartment) have been identified. Thereafter, we looked at the requirements set on the developers when building and what types of measures have been taken to reduce car dependence. When available, effects on car ownership, car usage, and mobility patterns were analyzed. Last, we studied motivation for lower MPR and what have been the success factors and challenges for these developments.

## 3. Results

The results are divided into two different sections related to the two main research questions. In the first part we summarize and analyze the evaluations of the developments. We look at the different type of developments based on size and parking norm, the different type of measures that have been implemented to reduce car dependency, the effects on mobility patterns and other insights from the evaluations. In the second part we analyze the quality of the evaluations. An overview of the evaluations can be found in the Supplemental Materials, Table S1.

### 3.1. Summary and Analysis of Evaluations

Figure 1 shows the residential buildings that have been evaluated and plot these based both on the size of the developments measured with the number of apartments and the parking norm, i.e. the number of car parking spaces/apartment.

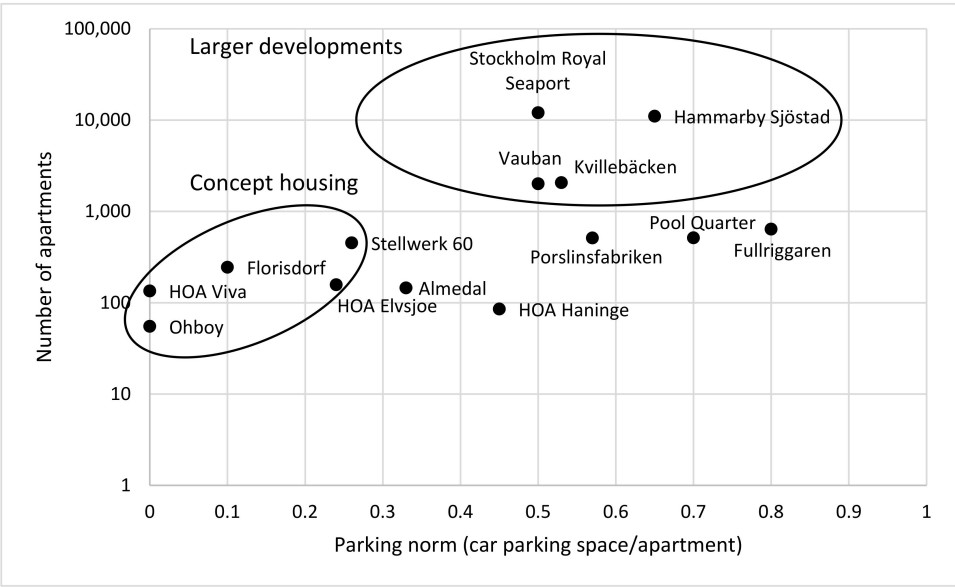

**Figure 1.** The studied developments plotted based on number of apartments and parking norm (car parking space/apartment). Two groups are identified: concept housing with parking norm below 0.3 car parking space/apartment and larger developments with at least 1000 apartments and parking norm above 0.5 car parking space/apartment.

Four case studies, Ohboy, Florisdorf, Stellwerk 60, and Viva, have a parking norm below 0.3 and were all marketed as "car-free". They can be seen as a type of "concept housing", where the absence of cars is prominent. These are smaller projects with less than 1000 apartments. While both Florisdorf and Stellwerk 60 have parking spaces, cars are not allowed in the residential area [20,21]. Parking spaces are limited to the outskirts of the area.

Four case studies, Stockholm Royal Seaport, Hammarby Sjöstad, Vauban, and Kvillebäcken, are larger residential areas and not individual buildings. The total number of apartments range from 2000 to 12,000 units. The parking norms are on the higher end of the studied developments (0.5-0.65) [22,23], but still lower than a traditional one parking space/apartment. They are or were developed during a longer period of time (5 to 26 years) and the whole residential area has environmental or sustainability programs or goals for other aspects than mobility and transport, such as energy efficiency or land use. These programs and goals are set up in collaboration between the municipality and the housing companies involved [24,25].

There are a variety of measures that have been implemented to reduce the dependency on privately owned vehicles and thus parking in the studied developments. These can roughly be divided into five different categories presented in Table 2.

**Table 2.** Type of measures offered to reduce the dependency on privately owned cars.

| Categories | Type of Services | Development |
|---|---|---|
| Car | Carsharing provided by housing company or another commercial actor | All |
| | Charging for electric vehicles | Stockholm Royal Seaport |
| | No car ownership binding contracts for tenants | Florsidorf, Stellwerk 60, Vauban |
| | High parking fees to cover actual costs | Florsidorf, Stellwerk 60, Vauban |
| Bike | Bikesharing | Hammarby Sjöstad, Ohboy, Stockholm Royal Seaport, Viva |
| | Cargo Bikes | Ohboy, Fullriggaren, Kvillebäcken, Viva |
| | High quality bike storage | Ohboy, Fullriggaren, Kvillebäcken, Viva, Embla Stockholm Royal Seaport |
| | Servicestation for bikes | Embla, Ohboy, Kvillebäcken, Viva |
| | Showers and changing rooms | Embla |
| | Discounts for bike purchases | Pool Quarter |
| Public transport | Discounted or free PT-cards during limited time | Ohboy, Elvsjoe, Haninge, Viva, Pool Quarter |
| | Real-time billboard for PT at entrance | Ohboy |
| Goods delivery | Boxes for packages | Ohboy, Viva |
| | Home delivery by bike | Sihlcity |
| General | Mobility management fund, i.e., part of the revenues from parking fees are set aside for investments that enhance sustainable mobility | Embla, Fullriggaren |

The Swedish and European developments differ in posed requirements for developers and tenants. For Swedish projects to achieve reduced parking requirements from the municipality, there are only obligations to provide some mobility services. The demands, from the municipality, are only set

on the developers and not on the residents and there is no compulsory follow up. In the European developments (except the UK example that is more similar to the Swedish cases) there are stronger demands on the residents that often have to sign contracts binding them not to own a vehicle. For commercial developments such as Sihlcity there are regulations on the amount of car trips to and from the area, with high fines if these are surpassed.

Most of the developments have been evaluated concerning the mobility patterns of the residents. In general, these patterns are more sustainable than in comparable areas (even if not all studies have a real control group to compare with—see section on quality of evaluations) and there are indications that there is a change of behavior regarding mobility. Workday trips are those that most often are done without a car even in the developments with a parking requirement of 0.5 parking space/apartment or higher. The car is mainly used for leisure trips and shopping trips.

In fourteen of the studied developments, private car ownership has been studied, and in the majority of the cases it is lower than in comparable areas (again the quality of this comparison is sometimes questionable). Some, like Almedal, still have a relatively high vehicle ownership (70%). For some of the studied developments, there is a moderate decrease in car use and car ownership. In Porslinsfabriken, for example, car use was reduced by 20% [26]. Why the number is not higher can partly be explained by good access to parking in the surrounding area. Even in Pool Quarter in the UK, vehicle ownership was reduced by about a quarter. The developments that are marketed as car-free in Figure 1 stand out since they have the lowest vehicle ownership and most sustainable mobility patterns. This is not that surprising, since the ambition is to be close to car free. In these developments, car trips are often substituted by biking or walking. When vehicles are used, they are from carsharing or car rental.

While the evaluations find that there is a positive effect regarding mobility patterns, some aspects should be kept in mind. First, the quality of the evaluation can, in many cases, be questioned. This also implies that the causality, especially the specific effect of lower parking requirements, is hard to establish. It is very plausible that these types of developments attract people that already have more sustainable mobility patterns. Several studies find that people tend to sort themselves in the neighborhood based on their travel mode preferences (see e.g., [27,28]). The issue of residential self-selection in relationship to mobility patterns is not always mentioned in the evaluations. Overall, those carried out by researchers more often discuss the issue of self-selection (see e.g., [18,29]). In the case of Florisdorf, 73% of the households did not previously own a car [30], while in Vauban, 81% of the car free households had previously owned a car [31]. In Viva, in a pre-survey, 66% of the respondents aimed at keeping their car despite there not being any residential parking.

Second, all the studied developments have good prerequisites for sustainable mobility, such as access to public transport, bike paths, central location, and good access to services. Previous research has shown that more transit-oriented development reduces the need for parking [32,33]. The interviewees in [26] point out that the location of Porslinsfabriken makes it easy to access what they need without a car. This implies that the main reason for the more sustainable mobility patterns is hard to establish. Johansson et al [29] find that several changes have occurred in the residents' life and that the change in vehicle ownership and mobility patterns is connected to bundled practices. However, access to parking or rather the lack of access to easy and cheap parking can be a contributing factor to go from car owning to being car free [29,34].

For some of the developments, access to more parking has been available in the surrounding area either in the form of a parking garage nearby or on-street parking. In Viva, those that chose to keep a car solved the parking problem by one of the following: renting or borrowing a parking space within walking distance, parking at a relative's property (often a parent or a child), or parking on the street [34]. Johansson et al [29] find similar solutions. It is thus important to look at the wider geographical area, the on-street parking availability, and not only the specific development project both in the planning phase and the evaluation phase of the project, since parking availability and pricing in

the surrounding area might affect the outcome and the development might also result in spill over parking [17].

In ten evaluations, the use of carsharing or other mobility services was also studied. Generally, membership in carsharing was higher than in the general population or in the surrounding area. In the studies that interviewed residents, carsharing was, for some respondents, still something new, and there was sometimes a lack of knowledge and confidence about these services [29,34]. Access to carsharing also implied an increase of car use for some individuals [29]. Johansson et al [29] also suggested looking at a larger area when developing the mobility services for the residents.

One important factor for a positive outcome of the developments, both when it comes to reducing car use and the satisfaction of all the involved parties, is a clear definition of roles between the actors. Legally binding contracts are needed to clarify responsibilities between developer, municipalities, and mobility service suppliers. For example, the contracts could cover which actor has the responsibility for providing and promoting the mobility services, and the business model used. The consequences of not fulfilling signed commitments (e.g., not providing an agreed number of shared cars) could also be brought up in the contract, as well as forms for terminating a mobility service before the contracted time ends in the case of a lack of interest from residents or potential users. In the projects in Germany, Austria, and Switzerland, there have also been contracts regulating car ownership of the residents.

Regarding specifically the mobility services provided, these should be included already in the planning process and permission and not be added ad hoc.

### 3.2. Quality of Evaluation

We now look at the quality of the evaluation taking into consideration who performed the evaluation, type of report produced, methods used, presence of a control group, if a before-and-after study was performed, and parameters studied (Table 3).

**Table 3.** Overview of projects and if the evaluation has a control group and if it is carried out by an independent researcher.

| Name of Development | Type of Building | Country | Control Group | Independent Researcher |
|---|---|---|---|---|
| Almedal | Residential | Sweden | No | Kind of |
| Elvsjoe | Residential | Sweden | No | Yes |
| Embla | Commercial offices | Sweden | No | No |
| Florisdorf | Residential | Austria | Yes | Yes |
| Fullriggaren | Residential | Sweden | "Yes" | Kind of |
| Hammarby Sjöstad | Residential | Sweden | "Yes" | Kind of |
| Haningen | Residential | Sweden | No | Yes |
| Ohboy | Residential (and hotel) | Sweden | No | No |
| Kvillebäcken | Residential | Sweden | No | No |
| Pool Quarter | Residential | UK | No | Yes |
| Porslinsfabriken | Residential | Sweden | No | Yes |
| Sihlcity | Commercial (shopping mall and offices) | Switzerland | No | No |
| Stellwerk 60 | Residential | Germany | "Yes" | Kind of |
| Stockholm Royal Seaport | Residential | Sweden | No | No |
| Vauban | Residential | Germany | Yes | Yes |
| Viva | Residential | Sweden | No | Yes |

Notes: "Yes" means that there is some comparison with e.g., the whole city or neighbouring area but no discussion or evaluation if this is appropriate. Kind of: master or bachelor thesis or researcher not focused on mobility

About half of the evaluations were carried out by independent researchers and most were reported in the grey literature. Seven of the case studies were evaluated by researchers with a focus on parking and mobility: Porslinsfabriken, Florisdorf, Vauban, Haningen, Elvsjoe, Viva, and Pool

Quarter [18,26,29–31,34]. Almedal Terrace, Fullriggaren, Hammarby Sjöstad, and Stellwerk 60 were evaluated in master or bachelor theses or by researchers with other focus than mobility. The remaining evaluations were carried out by the building companies, property owners, or the city.

Different methods have been used. The most common method for the evaluation was a survey to the tenants of the developments (11 out of 16 had done this) and in seven cases interviews were also performed to investigate attitudes or self-reported changes.

Most evaluations looked at mobility patterns in one way or another, i.e., to what extent different travel modes were used and for what purpose. In some cases, self-reported changes in mobility patterns were asked as well. Car ownership was reported in 14 out of the 16 evaluations. Vehicle registry data was used to determine car ownership levels in some of the evaluations. Two evaluations also looked at bike ownership and nine at carsharing membership. Attitudes towards carsharing and mobility were also reported. Three of the studies (Elvjsoe, Haninge, and Viva) used before-and-after surveys of the residents. However, the response rate was quite low and in the case of Viva it was not possible to determine if it was the same individuals responding. Self-reported changes and before-and-after surveys are susceptible to several forms of bias [35].

When it comes to having any form of control group, five out of sixteen evaluations compared parameters with a reference object. However, only in the case of Florisdorf and Vauban was the control group selected to make sure that location and demographics were similar. Income, employment, age, and household composition are important factors for car ownership and mobility patterns that need to be considered when selecting a control group. Evaluations of Stellwerk 60, Fullriggaren, and Hammarby Sjöstad compared parameters with reference areas such as a nearby city district or the city average. Stellwerk 60 and Fullriggaren parameters were compared with the average of the same the city district, without discussing differences in average apartment size, income levels, or demographics. Hammarby Sjöstad was compared both to a nearby city district and Stockholm municipality average values without commenting on the differences in demographics. Fullriggaren was also compared to a "reference district", but the characteristics of this district were not clearly explained, and neither were the criteria for selection.

The quality of evaluations regarding documentation vary. Non-academic evaluations sometimes lacked documentation of method used, population or sample size, or complete information on questionnaires. There was also a lack of consistency when evaluating mobility patterns, where different measures were used to describe travels. Stellwerk 60 modal share was based on total travel length by mode, whereas share of number of travels is a more common measure. Using different methods for assessing car ownership can result in different outcomes in the same city district, e.g., if registry data of privately-owned cars are used or if company cars are included. In Stockholm, car ownership rises between 2–20% depending on the district if company cars are included in the measure [36]. This has to be taken into consideration when comparing results, and when drawing conclusions. Another source of uncertainty or bias on reported results arises when, instead of comparing with a control group, data is collected from other sources, such as an annual city travel survey. The survey could have been carried out during a different time period, which in particular affects the share of bike trips.

There are not that many studies about best practices when it comes to studying real life transportation-related interventions. The literature is rather related to evaluations of social interventions (see e.g., [37]). Graham-Rowe et al. and Möser and Bamberg [38,39] reviewed transport policy measures and found that a large share of these have not been evaluated following strict qualitative criteria. The golden standard that they put forward are randomized control trials (RCT). Still, these are very hard to carry out in real-life experiments and may not always capture the underlying processes [40]. Melia [35] also questions if RCTs are always reliable when it comes to real life experiments. While they are likely to be more reliable when it comes to establishing if there is a cause-effect relationship, the magnitude of the effect may be harder to estimate [35].

Evaluations carried out by researchers rely in some cases on several sources of data, mixed-methods, or some kind of "triangulation" to get a better understanding of the change process (see e.g., [26,29,34]).

The interviews with several residents give a more in-depth understanding of the change process and the role of the different interventions, however, this method makes it harder to quantify the effect of, for example, a lower MPR for a new development. In Viva, a smartphone-based travel diary was tested afterwards (however not before) to better understand the mobility patterns. However, only a small share of the residents registered, thus opening for bias even in this case.

Quasi-experimental designs can be implemented in which there is a matched control group or some kind of cohort-analytic method where other groups are observed as well [38]. However, there will always be a selection bias. Even identifying and comparing with a control group can be challenging to implement. A control group could be selected from a larger sample, such as a general travel survey by statistical methods that reduce the selection bias [37]. One question is what parameters should guide the identification of a control group. It could be selected from a comparable location based on geodemographic profiling as well as proximity. Income and number of children are important, but also other factors, such as leisure activities and location of workplace, may influence mobility patterns. Næss [41] studies the effect of different control variables and finds that treating car ownership and attitudes to car travel as exogenous control variables that are not influenced by urban structure tend to underestimate the impact of residential location on mobility patterns.

More research on best practices of evaluation and selection of control group are needed. Another approach is to use several sources of data, such as GPS tracking or more automated collection of mobility patterns or traffic counts. Considering driving patterns in particular, naturalistic driving studies can be source of in-depth data and monitoring of driving behavior. GIS mapping could then be used to extract mobility patterns [42,43]. It would also be desirable to have more long-term evaluation to better understand, for example, how the adoption process changes, if vehicle ownership decreases over time and how the surrounding area is affected.

There are challenges in creating good evaluations in a naturalistic setting, and our results show that there is much room for improvement in the current evaluations and that these improvements are needed to increase the robustness of results, facilitate comparability of studies, and in the end, increase the knowledge of the effects of reduced parking requirements.

## 4. Future Outlook

As mentioned in the introduction, several Swedish municipalities have changed their parking policy in recent years. In a review of the parking policy of ten medium and large Swedish cities, the residential parking requirements now vary from 0 to 1.2 car parking space per regular new apartment that is built [44]. This is significantly lower compared to the situation ten to twenty years ago. For Gothenburg and Malmö (second and third cities in size), the parking requirements for apartment buildings now vary from 0.2 to 0.5 and from 0 to 0.4 car parking space per apartment in central areas. This means that the minimum parking requirement in most cases is lower than the level of car ownership in the cities. Further, most of the cities offer an interval for new housing at a specific location depending on the availability of mobility services. Generally, a reduction of the parking requirements of 10–25 percent is given depending on what mobility services are provided by the real estate owners [44].

This means that the Swedish projects studied in this paper are part of a bigger context and a more general development toward lower minimum parking requirements in Sweden. Even in other European cities parking management is being revised, see, for example, Kodransly and Hermann [15]. From a situation where low parking requirements have been introduced in smaller projects as the zero parking projects Ohboy in Malmö and BRF Viva in Gothenburg, there are now two other visible trends [45]. One is mentioned above that municipalities adopt lower and flexible parking requirements as a new standard for all new developments in a city. The other trend is that ambitions could be even higher for large areas and projects where municipalities and real estate owners work together. This implies that low parking developments are moving from niche projects to mainstream. In the coming years, these changes will be more visible since they will affect larger areas and a larger number of

residents compared to the situation today. In Stockholm, for example, 140,000 new apartments are planned during the next 15 years.

An example of a large project is Ulleråker in Uppsala, with 7000 new apartments with an average of 0.4 car parking spaces per apartment. Parking will be localized in parking houses at the edge of the area and the planning has a focus on biking, walking, accessibility to public transport, and a range of other mobility services such as carsharing [46].

Frihamnen in the central area of Gothenburg is one more big development area in Sweden with plans for 9000 apartments and 15,000 jobs in 2040. Regarding residential parking the ambition is to have a significantly lower parking norm compared to the requirements of today (0.2–0.5). Just as in Ulleråker, parking houses are planned at the outskirts of the area in combination with public transport and a wide range of other mobility services [47].

The scale of coming projects in combination with rising awareness of the question among different stakeholders will probably lead to new groups of residents in developments with lower parking requirements, i.e., not only residents that already have a "green" lifestyle with, for example, low car ownership. This implies that future evaluation will give more knowledge about the effects of lower parking requirements on the larger population. For these cases, acceptability of the measures will also be an important factor [48].

## 5. Conclusions

We reviewed 16 developments with minimum parking requirements ranging from 0 to 0.8 parking spaces per apartment. We found that the mobility patterns of individuals in the studied projects are more sustainable than in nearby areas. However, the causality of MPR and mobility is hard to establish. One of the reasons is due to the low quality of the majority of the evaluations. Additionally, all the studied projects had good prerequisites for sustainable mobility, such as access to public transport, a central location, mobility services, bike paths, and good access to services. Availability and price of parking in the surrounding area also affects the outcome of the project. It is thus important to take into consideration a larger area when evaluating the developments.

To reduce car dependence, many projects combine low MPR with mobility services such as carsharing. For this to be successful, requirements for MPR and mobility services should be included in the planning permission. Similarly, legally binding contracts are needed to clarify responsibilities between developer, municipalities, and mobility service suppliers.

Regarding the quality of the evaluation, we find that there is much room for improvement. More precisely, evaluations should be based on comparison with a control group selected based on relevant criteria and not just location. We acknowledge that this might not always be easy to implement but should still be the ambition. If this is not available and instead more general data is used, statistical methods should be used to artificially construct a control group and not just compare with the general population. Another factor that should be taken into consideration are seasonal differences that can affect, for example, biking frequencies.

Looking forward, we see that there are several even larger developments planned with lower parking requirements in Sweden. Many of these are being evaluated by researchers hopefully ensuring a better quality and therefore more robust knowledge of the connection between parking requirements, provision of mobility services, and mobility patterns.

**Supplementary Materials:** The following are available online at http://www.mdpi.com/2071-1050/12/5/1744/s1, Table S1.

**Author Contributions:** Conceptualization, F.S., C.H., Å.H., and A.R.; data curation, C.H. and Å.H.; formal analysis, F.S., C.H., Å.H, and A.R.; funding acquisition, F.S. and A.R.; Writing – original draft, F.S.; Writing – review and editing, F.S., C.H., Å.H., and A.R. All authors have read and agreed to the published version of the manuscript.

**Funding:** This research was funded by SWEDISH TRANSPORT ADMINISTRATION, grant/projekt MURF-GUTS, SWEDISH ENERGY AGENCY, grant number 44452-1, SWEDISH RESEARCH COUNCIL FORMAS, grant number 2017-01029

**Acknowledgments:** We want to thank Torunn Vikengren from Koucky and Partners for help with gathering of evaluations and analysis of these.

**Conflicts of Interest:** The authors declare no conflict of interest.

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
