# Peer review of "Review of the Effects of Developments with Low Parking Requirements"

_sustainability, doi:10.3390/su12051744_

Round 1

Reviewer 1 Report

Dear Authors

The paper is interesting , please check my comments and suggestions carefully

Increase spacing between characters:

Line 41  – “transport[4, 5, 9, 10].”

Line 43 – “cities[11].”

Line 52 – “clear[12]”

Line 56 – “[4, 5, 9, 14]has”

Line 222 – “(see e.g.[27, 28]).”

Line 249 – “individuals [29] .”

Line 317 – “(see e.g.[37]).”

Decrease spacing between characters:

Line 103 – “(eleven )”

Line 309 – “between 2-20 %”

At the beginning of the sentence starting with reference number add name before reference number, see two examples:

Line 38 - “[8] found, in Norway,”  – “Christiansen et al. [8] found, in Norway,”

Line 45 – “[12] lists” – “Shoup [12] lists”

Further sentences:

Line 66 – “[18] has studied”

Line 233 – “[29] find that”

Line 241 – “[29] find a similar”

Line 249 – “[29] also suggest”

Line 317 – “[38, 39] review”

Line 320 – “[35] also questions”

Line 339 – “[41] studies”

At the end of sentence, only in this case:

Line 366 – “see e.g. [15].” –“ see e.g. Kodransky [15].”

Line 89 – “such as carsharing, used?” Delete comma.

Is it possible to present five different categories (pointed below, page 5, line 169 - 191 ) in table? Car related Bike related Public transport related Goods delivery related General measures

It would be more clarified and readable

Table 2. What's your interpretation answer “kind of” in column 5 “Independent researcher”?

Can you explain it in short sentence under table like in case of “yes”?

Word “ideally” line 405 could be replaced by “more precise”.

Author Response

Thanks for the comments and edits. Please find the detailed response in the attachement

Reviewer 2 Report

What has surprised me is not finding any reference to naturalistic driving studies in your study. Several aspects related to this study are interesting to be evaluated using naturalistic driving data such as “travel behavior with regard to the mode of transport and trip purpose” [pages 29-31], determine “mobility patterns” [pages 36-37], etc and the effect of free workplace parking. These mobility patterns can be estimated by GPS tracking such as most naturalistic studies show.

A bit later, the authors refer to naturalistic settings, but they do not cite any referent work or study: “There are challenges in creating good evaluations in a naturalistic setting, and our results show that there is large room for improvement in current evaluations and that these improvements are needed to increase the robustness of results, facilitate comparability of studies and in the end increase the knowledge of the effects of reduced parking requirements” [pages 348-351].

For this reason, I will suggest to the authors to extend some parts of the study including some paragraphs with references to relevant studies related to Naturalistic Driving. This method enables us to collect exhaustive data related to a group of drivers under real driving conditions. It considers all the factors that intervene in driving performance and/or any decision related to that. Some studies related to naturalistic driving analyze how driving behavior and mobility patterns can be extracted from these kinds of data. I strongly recommend you add some reference to these works: “GIS mapping of driving behavior based on naturalistic driving data” (related to extracting mobility patterns and mapping driving behavior) and “Geo-referencing naturalistic driving data using a novel method based on vehicle speed” (related to GPS data for reconstructing mobility patterns).

Author Response

Thanks for the suggestion. The following sentences, with reference to the suggested papers has been added

“Considering driving patterns, in particular naturalistic driving studies can be source of in-depth data and monitoring of driving behavior. GIS mapping could then be used to extract mobility patterns [42, 43].